# In-Depth Investigation of Low-Abundance Proteins in Matured and Filling Stages Seeds of *Glycine max* Employing a Combination of Protamine Sulfate Precipitation and TMT-Based Quantitative Proteomic Analysis

**DOI:** 10.3390/cells9061517

**Published:** 2020-06-22

**Authors:** Cheol Woo Min, Joonho Park, Jin Woo Bae, Ganesh Kumar Agrawal, Randeep Rakwal, Youngsoo Kim, Pingfang Yang, Sun Tae Kim, Ravi Gupta

**Affiliations:** 1Department of Plant Bioscience, Pusan National University, Miryang 50463, Korea; min0685@naver.com; 2Interdisciplinary Program in Bioengineering, College of Engineering, Seoul National University, Seoul 03080, Korea; tryjpark@gmail.com (J.P.); biolab@snu.ac.kr (Y.K.); 3National Institute of Crop Science, Rural Development Administration, Wanju 55365, Korea; bjw0409@korea.kr; 4Research Laboratory for Biotechnology and Biochemistry (RLABB), GPO 13265, Kathmandu 44600, Nepal; gkagrawal123@gmail.com (G.K.A.); plantproteomics@gmail.com (R.R.); 5GRADE (Global Research Arch for Developing Education) Academy Private Limited, Adarsh Nagar-13, Birgunj 44300, Nepal; 6Faculty of Health and Sport Sciences, University of Tsukuba, 1-1-1Tennodai, Tsukuba 3058574, Japan; 7Key Laboratory of Plant Germplasm Enhancement and Specialty Agriculture, Wuhan Botanical Garden, Chinese Academy of Sciences, Wuhan 430074, China; yangpf@wbgcas.cn; 8Department of Botany, School of Chemical and Life Sciences, Jamia Hamdard, New Delhi 110062, India

**Keywords:** soybean, basic pH reverse phase, filter-aided sample preparation, low-abundance proteins, maxquant, protamine sulfate precipitation, perseus, tandem mass tags

## Abstract

Despite the significant technical advancements in mass spectrometry-based proteomics and bioinformatics resources, dynamic resolution of soybean seed proteome is still limited because of the high abundance of seed storage proteins (SSPs). These SSPs occupy a large proportion of the total seed protein and hinder the identification of low-abundance proteins. Here, we report a TMT-based quantitative proteome analysis of matured and filling stages seeds of high-protein (Saedanbaek) and low-protein (Daewon) soybean cultivars by application of a two-way pre-fractionation both at the levels of proteins (by PS) and peptides (by basic pH reverse phase chromatography). Interestingly, this approach led to the identification of more than 5900 proteins which is the highest number of proteins reported to date from soybean seeds. Comparative protein profiles of Saedanbaek and Daewon led to the identification of 2200 and 924 differential proteins in mature and filling stages seeds, respectively. Functional annotation of the differential proteins revealed enrichment of proteins related to major metabolism including amino acid, major carbohydrate, and lipid metabolism. In parallel, analysis of free amino acids and fatty acids in the filling stages showed higher contents of all the amino acids in the Saedanbaek while the fatty acids contents were found to be higher in the Daewon. Taken together, these results provide new insights into proteome changes during filling stages in soybean seeds. Moreover, results reported here also provide a framework for systemic and large-scale dissection of seed proteome for the seeds rich in SSPs by two-way pre-fractionation combined with TMT-based quantitative proteome analysis.

## 1. Introduction

Soybean seeds are considered as one of the most important crops because of their large-scale use as a dietary supplement to humans and livestock. As soybean seeds are rich in proteins, efforts have been put into investigating the soybean proteome in the last two decades, however, it remains poorly characterized due to the presence of high-abundance proteins (HAPs) which accounts for up to 75% of total protein content in soybean seeds [1,2,3]. Therefore, to increase the dynamic resolution of soybean seed proteome, various HAPs depletion techniques including protamine sulfate (PS) [4,5] and calcium [6] have been developed that facilitate the identification of low-abundance proteins on two-dimensional gel electrophoresis (2-DGE). These HAPs depletion methods seem to be effective and are broadly applied for the depletion of HAPs from different plants including broad bean, pea, wild soybean, and peanut, which contain their own specific HAPs [5]. Previously, the PS precipitation method allowed the identification of only a few specific low abundance proteins (LAPs) related to various functions in pea seeds such as transcription factors and regulators, kinases, transporter proteins, and various enzymes related to energy metabolism [7].

In the current era of high-throughput proteome analysis, the soybean seed proteome is still poorly dissected even after the development of methods for the depletion of HAPs. This is because of the primary utilization of a 2-DGE based proteomics approach which results in poor reproducibility, low sensitivity for LAPs, and poor resolution of specific proteins including small or large size of protein, acidic or basic proteins, and hydrophobic proteins [8,9]. Development of shotgun proteomic analysis techniques allows comparison of tandem mass spectra derived from peptide fragmentation that facilitates the relative (or absolute) quantification using the ion intensities or spectral counting and peptide labeling approaches [10]. Although several studies have used the labeling techniques to dig deeper into the proteomes of different organisms, some drawbacks still exist such as mass modification of labeled peptides reducing the accuracy of relative quantification, and different charge status produced in labeled peptides [11]. Thus, the tandem mass tags (TMTs) were presented as an alternative approach having the same features of other isotope labeling techniques, with mass tags of identical nominal mass and chemical structure, and that allowed up to 16 samples per MS run [9,12].

Previously, a TMT-based approach was utilized for the high-throughput proteome analysis of *M. truncatula* which resulted in the identification of more than 23,000 proteins along with 20,120 phosphorylation and 734 lysine acetylation sites [13]. Moreover, Lv et al. (2016) also performed proteomic analysis through a TMT labeling approach that led to the identification of a total of 6281 proteins and characterization of differentially modulated proteins during the seed aging condition of *T. aestivum* [14]. Although methodological advancements in various tools for shotgun proteomic analysis have facilitated the high-throughput proteome analysis of the legume seeds, the soybean seed proteome is still poorly characterized because of the high-abundance of SSPs. Therefore, in order to increase the proteome coverage and to identify the LAPs, we developed a pipeline utilizing PS-precipitation method together with TMT-based quantitative proteomic analysis of soybean seeds. Incorporation of PS-precipitation method (for protein fractionation) and basic-reverse phase chromatography (for peptide fractionation) resulted in a significant increase in the proteome coverage. The developed approach was then utilized for the comparative proteome analysis of two different varieties of soybean seeds during the seed filling stages to identify the proteins responsible for differential accumulation of proteins and oils in the two cultivars.

## 2. Materials and Methods

### 2.1. Experimental Design and Plant Materials

For mass spectrometry (MS) analysis of soybean seed proteins, two different sets of samples were used. We prepared a total six fractionated samples (*n* = 6) as a major group by novel protamine sulfate (PS) precipitation method for enrichment of low abundance proteins (LAPs) from two different cultivars of soybean seeds which contain high-protein (Saedanbaek) and low-protein (Daewon) content, respectively. Additionally, we have used a subgroup of *n* = 6 soybean seeds per seed filling stage from R5 to R7 stages. TMT data obtained were subjected to two-step normalization across the three technical replicates to overcome the source of variation that created from multiple labeling experiments.

Seeds of both the varieties (Daewon, and Saedanbaek) were sown in the experimental fields of the National Institute of Crop Science (NICS) Rural Development Administration (RDA) in Miryang, South Korea, in June. The soil was supplemented with a standard RDA N-P-K fertilizer (N-P-K = 3, 3, and 3.3 kg/10 acre). Seed filling stage samples were collected from R5 stage (30 days of seed filling stage) to R7 stage (63 days of seed filling stage) and matured seeds were harvested in October (average temperature, 23.5 ± 3.5 °C; average day length, 12 h 17 min) [15]. In addition, for evaluation of physiological changes, each harvested filling stage samples was tested for the dry weight and moisture content by drying in an oven at 65 °C for 48 h.

### 2.2. Confirmation of Proteins, Amino Acids, and Fatty Acids Content in Seed Filling Stage Sample

Total protein content in the soybean seeds was determined as described previously [16]. Briefly, using a protein analyzer (rapid N cube, Hanau, Germany) and high-speed vibrating sample mill (CMT T1-100, CMT Company Ltd., Tokyo, Japan), 50 mg of each sample was then wrapped in nitrogen-free paper and pressed to pellets. All the parameters were set to default. Glutamic acid (9.52% N) was used as a test standard, and a protein factor of 6.25 was used. Each sample was analyzed in triplicates.

Amino acids were analyzed using an HPLC (Dionex Ultimate 3000, Thermo Scientific, Waltham, MA, USA) as described previously [17,18]. Briefly, the derivatization of amino acids and alkyl amines with o-phthaldiadehyde (OPA) and 9-fluorenylmethyl chloro-formate (FMOC) was carried out online in the HPLC autosampler as described before [17,18]. The following volume of samples was injected to 5 μL borate buffer mixed with 1 μL of samples, 1 μL of OPA, 1 μL of FMOC reagent, and 32 μL of HPLC grade water. The analysis was performed using Inno C18 column (4.6 mm × 150 mm, 5 µm, Younjinbiochrom, Seongnam, Korea); the column temperature was set at 40 °C. The mobile phase comprised 40 mM sodium phosphate buffer, pH 7 (mobile phase A) and a mixture of water, ACN, and methanol (10:45:45, *v*/*v*%) (mobile phase B). The gradient elution was performed with 1.5 mL/min of flow rate as follows: 5% B for 3 min, gradually increase to 55% B within 21 min, 55% to 80% B within 1 min, sustained 80% B for 6 min, and linearly decrease to 5% B for 4 min. The excitation and emission of the fluorescence detector were set at 340 nm and 450 nm for OPA and 266 nm and 305 nm respectively for the detection of amino acids. The acquisition of data was processed by chromeleon 6.8 software (Thermo Scientific, Waltham, MA, USA).

Fatty acids (FAs) analysis were carried out as described previously [16]. Fatty acid methyl esters (FAMEs) were prepared by Soxhlet extraction, saponification, acid-catalyzed transesterification, and finally extraction of FAMEs in hexane. FAMEs were subsequently analyzed by capillary gas chromatography (column: 60 m × 0.25 μm I.D., 25.5 μm; flame ionization detected temperature, 210 °C; carrier gas, N2 at 1.0 mL/min; injector temperature, 210 °C; and oven temperature, programmed from 180 to 210 °C) using an Agilent 7890A capillary gas chromatograph. Quantitative data were calculated using the peak area ratio (% total FAs).

### 2.3. Protein Extraction and Digestion by Filter-Aided Sample Preparation

Total proteins from matured and seed filling stage soybean seeds were isolated using the protamine sulfate (PS) precipitation method with trichloroacetic acid (TCA)/acetone precipitation [5,19]. Briefly, for PS precipitation method, one gram of each seed powder was homogenized with 10 mL of ice-cold Tris-Mg/NP-40 extraction buffer (0.5 M Tris-HCl, pH 8.3, 2% (*v*/*v*) NP-40, 20 mM MgCl_2_) and centrifuged at 15,922× *g* for 10 min at 4 °C. The sequentially collected clear homogenate was incubated on ice for 30 min with 0.15% (*w*/*v*) PS stock solution. The extract was centrifuged at 15,922× *g* for 10 min at 4 °C to divide the PS-supernatant (PS-S) and PS-pellet (PS-P) fractions as described in previous reports [5]. Pellet fraction was dissolved in an equal volume of Tris-Mg/NP-40 extraction buffer and proteins were precipitated using TCA/acetone precipitation method. Finally, all three fractions that designated as total, PS-S, and PS-P were dissolved in 80% acetone containing 0.07% *β*-mercaptoethanol and stored −20 °C until further analysis.

Protein digestion was carried out using a filter-aided sample preparation (FASP) approach as described in a previous study [20]. Briefly, acetone-precipitated proteins (300 μg) were dissolved in 30 μL of denaturation buffer (4% sodium dodecyl sulfate (SDS) and 100 mM dithiothreitol (DTT) in 0.1 M tetraethylammonium tetrahydroborate (TEAB), pH 8.5). After sonication of the sample for 3 min and heating at 99 °C for 30 min, denatured proteins were loaded onto a 30-kDa spin filter (Merck Millipore, Darmstadt, Germany) and diluted with UA buffer (8 M urea in 0.1 M TEAB, pH 8.5) to a final volume of 300 μL. The buffer was washed and exchanged three times using 300 μL of UA buffer by centrifugation at 14,000× *g* for removal of SDS. After removing SDS from the samples, cysteine alkylation was accomplished through the addition of 200 μL of alkylation buffer (50 mM iodoacetamide (IAA), 8 M urea in 0.1 M TEAB, pH 8.5) for 1 h at room temperature in the dark. Then, the buffer was exchanged with UA buffer to TEAB buffer (50 mM TEAB, pH 8.5) in a spin filter unit. The protein was digested with trypsin (enzyme-to-substrate ratio (*w*/*w*) of 1:100) dissolved in 50 mM TEAB buffer containing 5% acetonitrile (ACN) at 37 °C overnight. After overnight digestion, the digested peptides were collected by centrifugation, and the filter device was rinsed with 50 mM TEAB and 50 mM NaCl. The peptide concentrations were measured using the Pierce Quantitative Fluorometric Peptide Assay (Thermo Scientific, Waltham, MA, USA) according to the manufacturer’s instructions. Moreover, we performed further sample preparation including peptides labeling and fractionation for MS analysis.

### 2.4. TMT Labeling, Desalting, and BPRP Peptide Fractionation Using Stage-Tip

TMT labeling of digest peptides was performed as described previously [21] and followed by the manufacturer’s instruction. Briefly, each TMT reagent (0.8 mg) was dissolved in 120 μL of anhydrous ACN, of which 25 μL was added to each channel of samples. Prior to incubation each sample sets, additional ACN was added as 30% of final concentration. After incubation at room temperature for 1 h, the reaction was quenched with hydroxylamine to a final concentration of 0.3% (*v*/*v*). Finally, combined samples at equal amounts across all samples and lyophilized the sample to near dryness and subjected to the desalting procedure. The pooled TMT-labeled peptides were desalted using the HLB OASIS column according to the manufacturer’s instruction [21,22]. Consequently, dried peptides were reconstituted in 200 μL of loading solution (15 mM Ammonium formate, 2% ACN) and loaded onto stage-tip prepared by packing C18 Empore disk membranes (3M, Bracknell, UK) at the bottom and POROS 20 R2 reversed phase resin into 200 μL yellow tip. Prior to load the peptides, the stage-tip was washed with 100% methanol, 100% ACN and equilibrated with loading solution. The peptide were loaded and 12 fractions were subsequently eluted with pH 10 buffer solution containing 5, 8, 11, 14, 17, 20, 23, 26, 29, 32, 35, 41, 44, 60, 80, and 100% ACN as described previous studies [23]. Finally, the 12 fractions were lyophilized in a vacuum centrifuge and stored at −80 °C until further LC-MS/MS analysis.

### 2.5. Q-Exactive MS Analysis

Obtained peptides were dissolved in solvent-A (water/ACN, 98:2 *v*/*v*; 0.1% formic acid) and separated by reversed-phase chromatography using a UHPLC Dionex UltiMate^®^ 3000 (Thermo Fisher Scientific, Waltham, MA, USA) instrument [24]. For trapping the sample, the UHPLC was equipped with Acclaim PepMap 100 trap column (100 μm × 2 cm, nanoViper C18, 5 μm, 100 Å) and subsequently washed with 98% solvent A for 6 min at a flow rate of 6 μL/min. The sample was continuously separated on an Acclaim PepMap 100 capillary column (75 μm × 15 cm, nanoViper C18, 3 μm, 100 Å) at a flow rate of 400 nL/min. The LC analytical gradient was run at 2% to 35% solvent B (100% ACN and 0.1% formic acid) over 90 min, then 35% to 95% over 10 min, followed by 90% solvent B for 5 min, and finally 5% solvent B for 15 min. Liquid chromatography-tandem mass spectrometry (LC-MS/MS) was coupled with an electrospray ionization source to the quadrupole-based mass spectrometer QExactive™ Orbitrap High-Resolution Mass Spectrometer (Thermo Fisher Scientific, Waltham, MA, USA). Resulting peptides were electro-sprayed through a coated silica emitted tip (Scientific Instrument Service, Amwell Township, NJ, USA) at an ion spray voltage of 2000 eV. The MS spectra were acquired at a resolution of 70,000 (200 *m*/*z*) in a mass range of 350–1650 *m*/*z*. The automatic gain control (AGC) target value was 3 × 10^6^ and the isolation window for MS/MS was 1.2 *m*/*z*. Eluted samples were used for MS/MS events (resolution of 35,000), measured in a data-dependent mode for the 15 most abundant peaks (Top15 method), in the high mass accuracy Orbitrap after ion activation/dissociation with Higher Energy C-trap Dissociation (HCD) at 32 collision energy in a 100–1650 *m*/*z* mass range [24]. The AGC target value for MS/MS was 2 × 10^5^. The maximum ion injection time for the survey scan and MS/MS scan was 30 ms and 120 ms, respectively. The mass spectrometry proteomics data have been deposited to the ProteomeXchange Consortium via the PRIDE partner repository with the dataset identifier PXD019155 [25].

### 2.6. TMT Data Analysis by MaxQuant and Perseus Software

The MaxQuant software (version 1.5.3.30) was used to apply for a database search as described previously [23,26,27]. All the three technical replicates of matured and seed filling stage samples were cross-referenced against the UniProt *Glycine max* database (75,674 entries, UP000008827, http://www.uniprot.org). TMT data processing of two different data was performed using default precursor mass tolerances set by the Andromeda search engine, which is set to 20 ppm for the first search and 4.5 ppm for the main search. Reporter mass tolerance has to set the minimum as 0.003 Da and the minimum reporter precursor ion fraction (PIF) was set to 0.5. The product mass tolerance was set to 0.5 Da and a maximum of two missed tryptic cleavage were allowed. Carbamidomethylation of cysteine residues and acetylation of lysine residues and oxidation of methionine residues were specified as fixed and variable modifications respectively. A reverse nonsense version of the original database was generated and used to determine the FDR which was set to 1% for peptide identifications. Statistical analysis was carried out using Perseus software (ver. 1.5.8.5) [28]. The normalization of reporter ion intensities was carried out by IRS_TMM methods as described previously [29,30]. Briefly, a two-step normalization method was applied through Bioconductor of R program in which sample loading (SL) normalization among the samples was conducted at first for corrections of sample loading errors using calculated mean from sum value of each column. As a next step, a second internal reference scaling (IRS) method with trimmed mean of M values (TMM) normalization was applied among the technical replicates represented by separate MS runs as introduced previously [29,30,31,32]. Missing values imputation was carried out from a normal distribution (width: 0.3, downshift: 1.8) using Perseus software [28]. Multiple sample test controlled by the Benjamini–Hochberg FDR threshold of 0.05, was applied to identify the significant differences in the protein abundance (≥1.5-fold change). The functional classification and pathway analysis were carried out using AgriGO v2.0 [33] and REVIGO [34] web-based software for GO enrichment analysis and MapMan software, respectively.

## 3. Results

### 3.1. TMT-Based Quantitative Proteomic Analysis of PS-Fractionated Mature Soybean Seed Samples

To analyze the fractionation efficiency of soybean seed proteins by PS [1,2], protein profiles of the total, PS-supernatant (PS-S), and PS-precipitant (PS-P) fractions were generated using a TMT-based proteomics approach (Figure 1A). Proteins from these three fractions were isolated from Daewon (D, low protein) and Saedanbaek (S, high protein) varieties of soybean and subsequent SDS-PAGE analysis showed clear separation and effective depletion of major SSPs in PS-P fraction, similar to the protein profiles of PS-P fractions reported previously [5] (Appendix A). For TMT-based proteome analysis, isolated proteins from all the fractions were subjected to in-solution trypsin digestion by the FASP method [20] and digested peptides from 3 biological replicates of 6 samples were labeled with TMT 6-plex kit (Figure 1A). To reduce the complexity of the multiplex labeling sample mixtures, TMT-labeled peptides were further fractionated into 12 fractions by basic pH reversed-phase (BPRP) using an in-house developed stage-tips [22]. Altogether, the sequential LC-MS/MS analysis led to the identification of 5199 protein groups (Figure 1A and Appendix A). The average sequence coverage for these identified proteins from three fractions was 23.9, 23.9, and 23.8, respectively, with an average score of 45.73 (Figure 1A and Appendix A).

In addition, to investigating the reproducibility of technical replicates, cross-correlation examinations were also performed where technical triplicates of the same samples showed outstanding consistency with an average R^2^ value of 0.996 (Appendix A). Further multiple normalizations applied showed improvement of the coefficient of variation (CV) values of each channel reporter ion intensities from 24.9% to 8.7% (Appendix A). This approach showed increased resolution of the seed proteins as observed by the identification of 88.4% (2106) higher unique proteins as compared to the previous studies [5,6,15,16,26] whereas one-way fractionation using PS and calcium combined with label-free and 2-DGE approach identified comparatively less number of identification (79 and 99 respectively) than this study (Figure 1B,C).

Of the total 5199 identified protein groups, 4074 proteins showing more than 70% valid intensity values within three replicates of each sample were selected for the downstream analysis (Appendix A). Student’s *t*-test controlled by a Benjamini–Hochberg FDR threshold of 0.05 with a fold change of more than 1.5 was applied to identify statistically significant proteins among the three fractions. A total of 2200 differentially modulated proteins were identified which were segregated into two major clusters of 1011 (cluster_1) and 1189 (cluster_2) proteins showing differential modulation in PS-S and PS-P fractions, respectively (Appendix A).

### 3.2. Functional Classification of Identified Proteins by PS Fractionation Method

GO enrichment analysis using AgriGO v2.0 [33] and REVIGO [34] was carried out for the functional annotation of the identified proteins (Appendix A). In the case of molecular function category, proteins related to the catalytic activity (562 proteins) and structural molecular activity (136 proteins) were dominated in cluster_1 and cluster_2, respectively (Appendix A). Of these 562 proteins in cluster_1, 21.4% were associated with hydrolase activity, 25% with oxidoreductase activity, 8.5% with peptidase activity, 8.5% isomerase activity, 7.1% with lyase activity, 5.2% with ligase activity, among others. In contrast, 136 proteins of cluster_2 were mainly associated with the structural constituent of ribosome (96.7%), structural constituent of the nuclear pore (3.7%), and others (3.7%) categories (Appendix A). Further MapMan pathway analysis revealed that 562 proteins of cluster_1 were mainly related to the amino acid metabolism, major CHO metabolism, and lipid metabolism, in metabolism overview category and protein degradation, redox, post-translation modification, signaling, and among others in cell function category (Figure 2A,B). The proteins of cluster_2 were mainly related to protein synthesis, and cell organization and transport, among others (Appendix A). In order to get further functional insights and visualization of fold changes, a protein-protein interaction map was generated using the STRING database with Cytoscape software based on major metabolism and cell function of enriched proteins in both PS-S and PS-P fractions. Particularly, protein-protein interaction analysis led to the identification of interacting partners of each protein involved in various metabolic functions including photosynthesis, major CHO metabolism, glycolysis, glucogenesis, TCA cycle, and mitochondrial electron transport (Figure 2C). Furthermore, several proteins were also found to be associated with protein degradation, protein amino acid activation, and protein targeting (Figure 2D). The proteins enriched in PS-P fraction (cluster_2) were exclusively related to various ribosomal proteins (data not shown). As these results showed clear depletion of SSPs in the PS-P fraction and subsequent enrichment of LAPs in the PS-S fraction (Figure 2) and increased the dynamic resolution of soybean seed proteome, the same approach was utilized for the comparative proteome analysis of high- (Saedanbaek) and low- (Daewon) protein varieties of soybean during seed filling stages was generated. However, prior to proteomic analysis, we carried out validation of physiological properties by measurement of dry weight, moisture, protein, fatty acids (FAs), and free amino acids (FAAs) contents before TMT based proteome analysis for the different profiles of the proteome during the seed filling stage.

### 3.3. Physiological Validation and Free Amino Acids Analysis Using Seed Filling Stage Samples

Field-grown samples were harvested at six-time points starting from the R5 stage (30 days) to R7 stage (63 days) of both Daewon (D) and Saedanbaek (S) varieties, and dry weight and moisture contents were measured (Figure 3A,B). The Daewon and Saedanbaek seeds showed a gradual increase in dry weight with seed desiccation. In particular, the Daewon variety revealed around 25% higher dry weight than Saedanbaek from 51 days to 63 days of seed filling, whereas the moisture content was almost similar during all the stages of seed filling (Figure 3B). In the case of protein and oil contents, an opposite trend was observed. The protein content was 0.95%, 3.12%, 7.65%, 21.75%, 37.42%, and 37.4% in Daewon seed and 1.85%, 4.17%, 13.73%, 25.78%, 42.59%, and 45.28% in Saedanbaek seed, respectively, during the seed filling stages, suggesting that Saedanbaek variety contains around 17% higher protein than Daewon in the R7 stage (Figure 3C). In addition to the protein profile, total FA levels were also analyzed in both varieties including palmitic acid (C16:0), stearic acid (C18:0), oleic acid (C18:1), linoleic acid (C18:2), and linolenic acid (C18:3) (Figure 3D). Among all the FAs analyzed, the content of linoleic acid was the highest, and was found to be two times higher in Daewon than Saedanbaek. In case of Daewon, contents of linoleic acid were recorded as 15.55, 43.88, 63.48, 86.71, and 92.98 (mg/g) while it was 14.96, 33.68, 44.88, 55.06, 58.22, and 53.50 (mg/g) in Saedanbaek (Figure 3D). The contents of other FAs including palmitic acid, stearic acid, oleic acid, and linolenic acid were found to be higher in Daewon as compared to Saedanbaek (Figure 3D). Further, analysis of the 44, 51, and 58 days of seed filling samples showed a rapid accumulation of proteins. Moreover, the overall concentration of majority of the essential and non-essential amino acids were higher in Saedanbaek as compared to Daewon (Figure 4A,B). In particular, asparagine and glutamine were 5.2- and 6.7-fold higher in Saedanbaek than Daewon. In addition, the contents of essential amino acids including histidine, leucine, isoleucine, lysine, methionine, phenylalanine, threonine, and valine were also found to be higher in Saedanbaek than Daewon. However, the concentration of tryptophan was 1.7-fold higher in Daewon as compared to Saedanbaek variety (Figure 4A,B).

### 3.4. Identifying the Proteome Changes between Two Varieties and Functional Annotation of Seed Filling Stage Samples

To investigate the alteration of proteins that contribute to the different physiological characteristics of two soybean varieties during 44, 51, and 58 days of seed filling stages, the proteins were isolated using the PS precipitation method (Appendix A) and analyzed using the same approach as described above. This approach led to the identification of 54,077 peptides and 23,351 unique peptides, matching to 5918 protein groups (Figure 5A and Appendix A). The average score for these identified proteins was 38.93 with average sequence coverage of 24% (Appendix A). Out of 5918 protein groups, 5663 and 3350 showed more than one unique peptide and 70% valid reporter ion intensities, respectively among three replicates of each sample (Figure 5A). Normalized data sets by the same method as specified above showed improvement of CV values from 34.3% to 4.2% in raw data and IRS_TMM normalized data, respectively (Figure 5B and Appendix A). Pearson’s correlation coefficient values of triplicates of the same sample showed a high degree of correlation and ranged from 0.96 to 0.99 (Figure 5C). The PCA plot analysis revealed that 44 vs. 58 days and 51 vs. 58 days of seed filling sample in both Daewon and Saedanbaek varieties were separated at the PC1 accounting for a maximum 70.8% variation and PC2 accounting for 9.7% variation, respectively (Figure 5D). Further, the reproducibility of technical replicates was measured by cross-correlation examinations that showed an average R^2^ value of 0.986 (Appendix A).

Student’s *t*-test controlled by a Benjamini–Hochberg FDR < 0.05 with more than 1.5-fold change was applied to identify the significantly modulated proteins among different stages of seed filling, resulting in the identification of 924 differential modulated proteins (Figure 6 and Appendix A). Of these 924 differentially modulated proteins, 63, 413, and 448 proteins showed increased and decreased abundances in cluster_1, 2, and 3, respectively (Figure 6A,B). In addition, further, GO enrichment analysis showed the functional classification of proteins involved in each cluster (Figure 6C and Appendix A). Particularly, proteins involved in cluster_2 showed up-regulation of proteins associated with photosynthesis (GO:0015979), organonitrogen compound biosynthetic process (GO:1901566), and cellular component organization (GO:0071840) whereas increased abundance in cluster_3 mainly dominated by proteins related to the catabolic process (GO:0009056). In addition, MapMan pathway analysis of differential proteins showed an up-regulation of photosynthesis (26.8%), cell wall modification (14.9%), secondary metabolism (13.1%), and among others in cluster_2 whereas proteins involved in cluster_3 were mainly associated with cell wall metabolism (18.4%), major carbohydrate metabolism (18.4%), glycolysis (11.5%), and among others in metabolism overview category. respectively (Appendix A). In the cell function overview category, the majority of proteins in cluster_2 were related to protein synthesis (24.7%) while proteins in cluster_3 showed majorly associated with protein degradation (24.9%) and developmental process (13.7%) (Appendix A).

## 4. Discussion

The SSPs in soybean seeds majorly include different subunits of *β*-conglycinin and glycinin. These SSPs act as storage reserves for nitrogen, carbon, and sulfur serving as energy sources during the development and germination of seeds [35]. However, a higher proportion of these SSPs limits the resolution of the proteome in both 2-DGE and shotgun proteomic analyses, therefore, multiple methods have been developed for the efficient and selective depletion of these SSPs using PS [5], calcium [6], and isopropanol [36]. A previous study suggested that pre-fractionation during protein extraction leads to sufficiently decreased contents of major SSPs from total proteome [2]. However, to define the molecular and physiological mechanisms of LAPs in soybean seed underlying the contribution of HAPs, a depletion method is currently elusive. A previous study has shown that one-way fractionation using PS precipitation combined with label-free analysis resulted in the identification of a few LAPs involved in various metabolic pathways, however, the number of identified proteins was almost similar to 2-DGE analysis (Figure 1B,C) [6,15,16,26]. Therefore, in this study, we carried out two-way fractionation using PS precipitation and BPRP chromatography combined with TMT-based proteome analysis for an in-depth understanding of the proteomic changes, particularly focusing on the LAPs. At first, a validation of the two-way fractionation with TMT-based analysis was carried out in matured seeds. Results revealed that proteins were clearly separated in total, PS-S, and PS-P fractions, and resulted in the identification of 2200 proteins. In particular, 2106 proteins were uniquely identified through this (current study) approach over the previous one-way fractionation approaches analyzed by label-free and 2-DGE [5,6,16,26,35]. Therefore, we generated subgroups for validation of further proteome alterations followed by the analysis of LAPs enriched PS-S fractions from seed filling stages of two soybean cultivars for comparative proteome analysis. Consequently, additional pre-fractionation of TMT-labeled peptides into 12 fractions by BPRP chromatography further improved the identification of LAPs and resulted in the identification of 924 significantly modulated proteins (FDR < 0.05, fold change > 1.5) from seed filling stage samples. Further, functional annotation of the differential proteins revealed accumulation of LAPs related to various metabolic pathways in PS-S fraction of filling stage seeds including protein degradation, protein synthesis, photosynthesis, and major carbohydrate metabolism, among others.

### 4.1. Enrichment of Proteins Participating in Major Metabolism in Seeds

Results obtained here using the HAPs depletion method combined with TMT-based proteomic analysis showed clearly separated protein profiles of both PS-S and PS-P fractions. The presence of SSPs in the seeds presents a major limitation in the dynamic resolution of the seed proteome [36,37]. Peptide pre-fractionation techniques such as BPRP or SCX reduces the complexity of pooled peptides and increases the protein coverage in the shotgun proteomic analysis [2], nevertheless, HAPs lead to masking or suppression of LAPs during 2-DGE and shotgun proteomic analyses [38]. Therefore, many researchers have developed new methods for depletion of major SSPs from diverse plant seeds using isopropanol [36], acid-SDS [39], chloroform-assisted phenol extraction (CAPE) [40], ethanol precipitation method (EPM) [41], PS [5], calcium [6], PEG [42], concanavalin-A/lectin affinity chromatography [43], and combinatorial peptide ligand library (CPLL) [37,44]. Following the use of these methods, significant depletion of SSPs was observed on 2-DGE from soybean seeds, broad bean, peanut, pea [5,6], lettuce [45], and maize [40]. A comparison between the fractionated and unfractionated samples strongly indicated that the number of identified SSPs was around 20% less in the fractionated sample than the unfractionated one [2]. The characterization of molecular functions with identified proteins was particularly represented by the up-regulation of various metabolism events in PS-S fraction including photosynthesis (such as cytochrome b6; cyt b6, Ribulose-phosphate 3-epimerase; RPE, triosephosphate isomerase; TPI, coproporphyrinogen III oxidase; CPOX, phosphoribulokinase; PRK, ribulose bisphosphate carboxylase; RuBisCO), major carbohydrate metabolism (such as starch synthase, sucrose synthase;SuSy, glyceraldehyde-3-phosphate dehydrogenase;G3PDH, pyruvate kinase;PK, enolase), TCA cycle (such as pyruvate dehydrogenase;PDH, succinate CoA ligase, malate dehydrogenase; MDH, isocitrate dehydrogenase;IDH, succinate dehydrogenase; SDH), amino acid metabolism (such as asparagine synthetase, homoserine dehydrogenase, cysteine synthase, arginase, urease, methionine S-methyltransferase), and secondary metabolism. Furthermore, enrichment of LAPs led to the identification of proteins involved in diverse cell functions, particularly signaling components such as RAS and RAB proteins, a family of small GTPase and G proteins that regulate signal transduction, cell proliferation, cytoskeletal organization, and intracellular membrane trafficking [46]. Besides, we observed the accumulation of 97 proteins mainly related to the protein homeostasis which cooperatively acts to protein synthesis, folding, and degradation for maintain the protein pools in the living organism for regulation and maintenance of cell function, organismal health, and adapt to environmental challenges [47]. However, the enrichment of HAPs in PS-P fraction led to the accumulation of the proteins mainly involved in ribosomal proteins and SSPs.

### 4.2. Changes in the Free Amino Acids Profile Have a Positive Correlation with Protein Accumulation of Soybean Seeds during Seed Filling Stages

Remobilization and transport of various metabolites including amino acids are required for the growing embryo during seed filling [48]. Seed reserves such as protein, oil, and starch are catabolized by the vegetative organs to supply amino acid and/or sucrose to the developing embryos [49]. A pool of FAAs has important roles in various metabolic processes and serves as the main precursors for diverse primary and secondary metabolites, including organic acids, amino acids, and other metabolites during seed filling [50].

Results obtained by FAAs analysis have shown that most FAAs were higher in the Saedanbaek variety including essential and non-essential amino acids. Interestingly, our FAAs analysis revealed the concentration of glutamine and asparagine were 5.2- and 6.7- fold higher in Saedanbaek than Daewon variety, respectively. Generally, inorganic and organic nitrogen sources are absorbed (or fixed) from maternal vegetative tissues such as leaves and roots. Initially, it reassimilates into glutamine through glutamine and glutamate synthase pathway from ammonia derived from asparagine [51] and that remobilized nitrogen source is mainly served as glutamine and asparagine from maternal organ to the developing embryo [49]. Interestingly, a previous study has demonstrated that quantitative trait locus (QTLs) on chromosome 20 and 15 (LGs I and E) influencing seed protein concentration positively correlates with a higher level of free asparagine in developing seeds [52]. In addition, it was shown that a higher level of asparagine synthase protein in seed coat might be contributing to nitrogenous assimilation to supply amino acid majorly provided as asparagine in developing embryo via the biosynthetic pathway of asparagine [52]. In particular, we observed that the amount of asparagine and glutamine mainly exhibited a decrease in both varieties from 5907 to 557 (mg/kg), 5625 to 2933 (mg/kg) and 275 to 39 (mg/kg), 1253 to 143 (mg/kg) in Daewon and Saedanbaek, respectively as compared to the 44 and 58 days of seed filling. However, the amount of both FAAs showed was comparatively maintained at a higher level in the Saedanbaek variety. A previous study showed decreased levels of FAAs in Arabidopsis and Canola seeds are accompanied by an increase in total amino acids (TAAs) which implies a few major FAAs incorporated into storage proteins during development of seed through catabolic pathways [53,54]. Although it was suggested that a higher amount of those FAAs inferring as potential availability to synthesize storage proteins, it might help to interpret a key trait for improvement of soybean value [50].

On the other hand, the concentrations of arginine and methionine were found to be higher in the Saedanbaek seed than all three filling stages of the Daewon seed, which accounted for a maximum of 3.9- and 2.2-fold increase in arginine and methionine, respectively. Likewise, glutamine and asparagine are major products of nitrogenous assimilation, and arginine also serves as a major storage form of organic nitrogen accounting for around 50% of the total nitrogen reserve in the total FAAs pool of developing embryo of soybean [55,56]. Besides, these amino acids have a role as an amino acid for protein synthesis and the precursor of polyamine and nitric oxide during the developmental procedure of seeds [55,56]. A previous study suggested that the balance of energy status and efficiency of nitrogen assimilation via glutamate and glutamine synthase pathway for nitrogen/carbon availability is important for the biosynthesis of arginine [57]. In addition, the sulfur-containing amino acid, methionine, is known as proteinogenic amino acid and plays a major role in diverse metabolic pathways especially being incorporated into the storage protein synthesis steps [58]. A higher concentration of methionine led to alteration of the composition of storage proteins, where in particular the 12S-albumins and 2S-globulins presented higher amounts in transgenic Arabidopsis seeds [59]. In addition, Song et al. (2013) demonstrated that transgenic soybean with overexpression of Arabidopsis cystathionine γ-synthase (CGS) showed significant accumulation of methionine than the wild-type seeds and up to a 3.5-fold increase of FAAs in ZD24 cultivar of soybean seeds was seen [60].

Consequently, a positive correlation between proteins and FAAs including asparagine, glutamine, arginine, and methionine indicated that nitrogen (or carbon) uptake (or transport) from maternal organs and nitrogenous assimilation plays a crucial role for regulating protein accumulation during seed filling stages. Particularly, the maintenance level of free arginine in developing seeds is mainly served as essential metabolites for the cellular and developmental processes.

### 4.3. Differential Regulation of Photosynthesis and a Major Cho Metabolism between Two Varieties of Soybean Seeds during Filling Stages

The present study’s experimental design allowed for the identification of various proteins involved in major metabolism during seed filling stages. These were mainly related to photosynthesis and CHO metabolism. Particularly, the differential abundance of 2 isoforms of photosystem (PS) I subunit, 10 isoforms of PS II subunit, 13 isoforms of light-harvesting complex (LHC), and 2 isoforms of cytochrome b6 involved in photosynthesis light reaction of green plant seeds was observed (Figure 7). The accumulation of storage compounds such as protein and oil requires a considerable amount of energy, however, the limitation of energy in the inner part of developing seeds suffered due to a shortage of oxygen and light penetration [61]. Parallel roles of photosynthesis during developmental stages of dicotyledonous green seeds described in the previous report suggested a reduction of photosynthetic activity contributed to the energy status of seed [62]. Besides, few enzymes involved in major metabolic pathways including SuSy, G3PDH, PK, PDH, citrate synthase (CS), IDH, phosphoenolpyruvate carboxylase (PEPC), and phosphoenolpyruvate carboxykinase (PEPCK) were also found to be differentially modulated (Figure 7). Particularly, PEPCK proteins showed increased abundance in the Saedanbaek variety of three filling stages. The PEPCK is an enzyme that catalyzes the ATP-dependent decarboxylation of organic acids, oxaloacetate, to form phosphoenolpyruvate (PEP) [63]. A previous study suggests a regulatory role of PEPCK in gluconeogenesis, by acting as a supplier of carbohydrates from lipids and proteins in post-germinated seeds [64]. On the other hand, the PEPCK enzymes in developing seeds mainly contributed to the synthesis of storage products such as lipids and proteins in grape [65] and pea seed [66]. In particular, Adriana et al. (2007) suggested the role of PEPCK in developing pea seeds that abundance of PEPCK has mainly affected to storage protein synthesis and assimilation of various metabolites such as sucrose, amino acids, and amides in developing pea seeds [66].

### 4.4. Differential Regulation of Protein Degradation during Seed Filling Stages

The proteomic comparison between Saedanbaek and Daewon varieties showed differential regulation of proteins related to protein degradation and synthesis. Here, we identified various proteases including 9 isoforms of subtilisin-like serine protease, 8 isoforms of cysteine protease, 8 isoforms of aspartate protease, 8 isoforms of serine protease, and a few hydrolase enzymes that showed almost equivalent profiles in both the varieties (Figure 7). A higher amount of various protease enzymes including aspartic protease, serine protease, aminopeptidase, proteasome, and subtilisin-like serine protease was found in developing seed for remobilization of endogenous nitrogenous products such as amino acids and/or proteins, which preferentially accumulated in both endosperm and seed coat and fuel as the synthesis of storage proteins in the embryo [67]. Further confirmation of remobilization of endogenous nitrogen was carried out in a previous study where the seed cultured in the absence of exogenous nitrogen showed remobilization of nitrogen compounds available in seed surrounding tissues in in-vitro condition [68]. In addition, our results showed the presence of ubiquitin-related proteins including 3 isoforms of ubiquitin-protein ligase (E3), ubiquitin-associated (UBA) domain-containing protein, ubiquitin-like domain (ULD) containing protein, and 30 isoforms of 26S proteasome subunit (Figure 7). The ubiquitin-proteasome system (UPS) mediated protein degradation regulates almost all the aspects of numerous physiological events during plant growth, development including embryogenesis, hormone signaling, and senescence by interfering with the essential component of these pathways [69]. In particular, UPS associated pathway reveals storage component degradation for seed germination [70], regulation of hormonal signaling during various abiotic stress condition [69], and the potential role for plant-microbe interaction such as control the function of pattern recognition receptors (PRRs) by E3 ligase of ubiquitin [69,71]. Moreover, UPS-mediated diverse hormonal signaling including jasmonic acid, ethylene, gibberellic acid, and others showed multi-functional aspects for promoting seed development via downstream signaling of phytohormone [72]; however, the differential modulation of UPS-associated proteins during the seed filling stages is still not fully understood.

## 5. Concluding Remarks

In this manuscript, we present a pipeline for the high-throughput proteome analysis of soybean seeds using a two-way fractionation method combined with a TMT-based quantitative proteome analysis. Application of PS-precipitation method resulted in significant depletion of SSPs in the PS-P fraction while enriching the LAPs in the PS-S fraction. On the other hand, TMT-based quantitative proteome analysis together with basic reverse phase fractionation of the pooled peptides further increased the dynamic resolution of soybean seed proteome, leading to the identification of more than 5900 proteins which is almost five-fold higher than previously reported. Functional annotation of the specific proteins showed proteins related to the photosynthesis, CHO metabolism, glycolysis, glucogenesis, and TCA cycle were specifically enriched in the PS-S fraction while those related to the ribosomal proteins were enriched in the PS-P fraction in matured soybean seeds. Further comparative analysis of the filling stages seeds from two varieties showed differential regulation of various proteins associated with photosynthesis, and major CHO metabolism, among others. Taken together, our results not only provide a workflow for the high-throughput proteome analysis of soybean or other leguminous seeds rich in HAPs but also provide a list of protein candidates involved in the differential regulation of proteins and oils in two soybean varieties.

## Figures and Tables

**Figure 1 cells-09-01517-f001:**
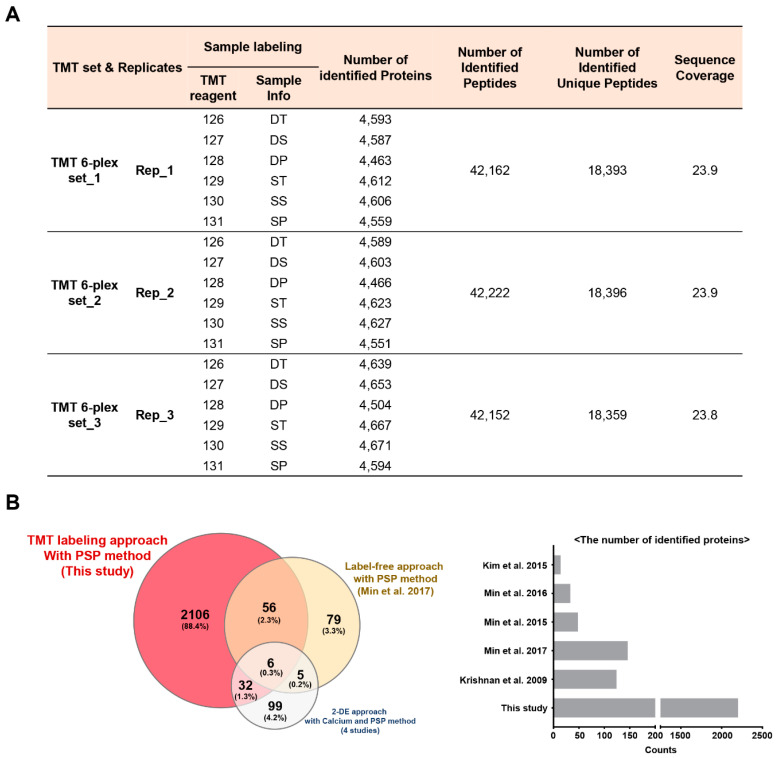
Proteomic analysis of soybean seeds extracted by protamine sulfate precipitation method combined with tandem mass tags (TMT)-based quantitative approach. (**A**) Each sample labeled with different TMT reagents of the 6-plex kit as listed in the table. (**B**) A recently published soybean proteome data analyzed with the pre-fractionation method combined with 2-DGE and label-free was compared to our TMT-based quantitative proteome analysis result.

**Figure 2 cells-09-01517-f002:**
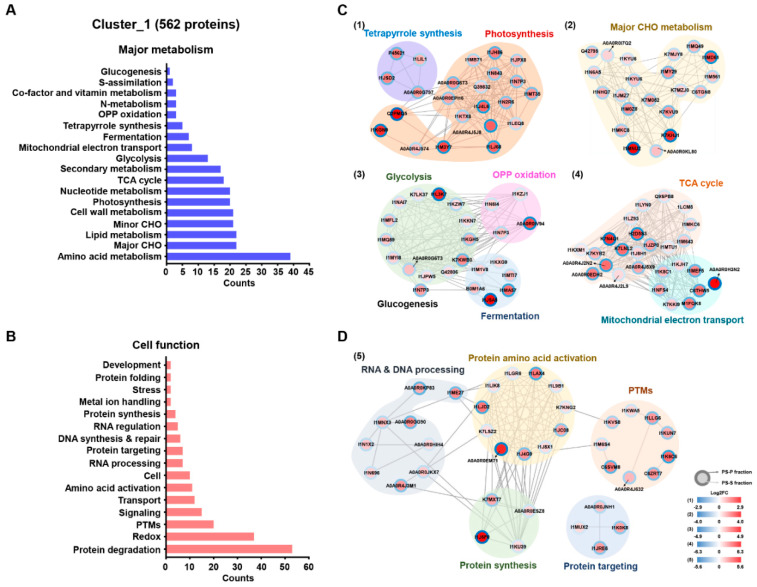
Functional classification of the identified and enriched proteins in cluster_1 using MapMan software. Differentially modulated proteins (with more than 1.5-fold change differences among each sample) were distinctly characterized as major metabolism overview (**A**) and cell function overview (**B**) categories. The abundance level of significant proteins involved in metabolism overview (**C**) and cell function overview (**D**) derived from the MapMan analysis showing the functional network with associated proteins. The fold change differences of individual proteins were visualized by colored nodes (red: increased abundance, blue: decreased abundance).

**Figure 3 cells-09-01517-f003:**
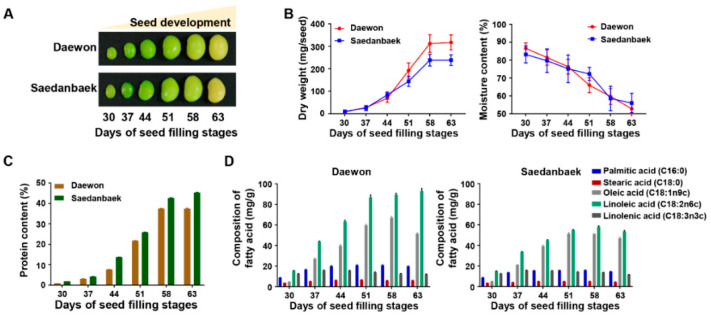
Physiological validation of filling stage samples. The morphological changes (**A**), dry weight, and moisture content (**B**) of Daewon and Saedanbaek varieties of soybean seeds during seed filling stages. A bar chart showing variations in the protein (**C**) and fatty acids contents (**D**) in Daewon and Saedanbaek varieties during seed filling stages.

**Figure 4 cells-09-01517-f004:**
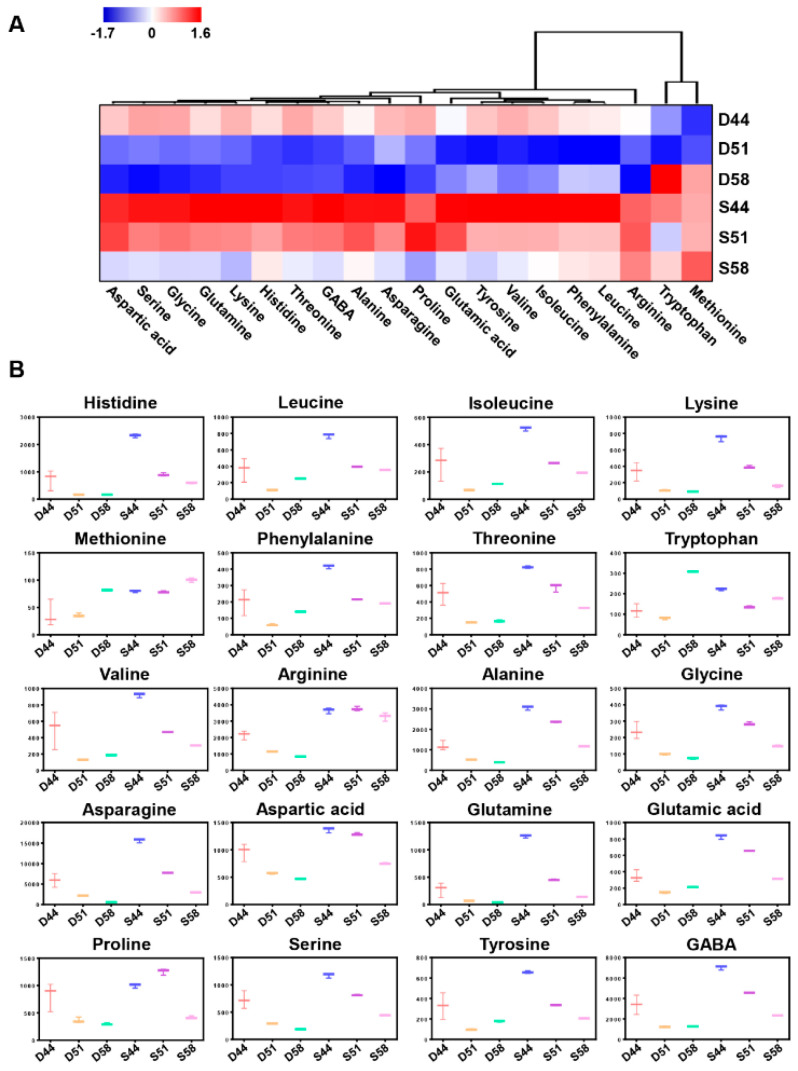
Relative quantification of free amino acids in filling stage seeds. (**A**) HCA clustering of each amino acid by MeV software, where red and blue colors indicate up- and down-regulation, respectively. (**B**) Box plots of 20 amino acids represent differential contents of nine essential and eleven nonessential amino acids. Error bars indicate standard deviations obtained by three replicates of the same sample.

**Figure 5 cells-09-01517-f005:**
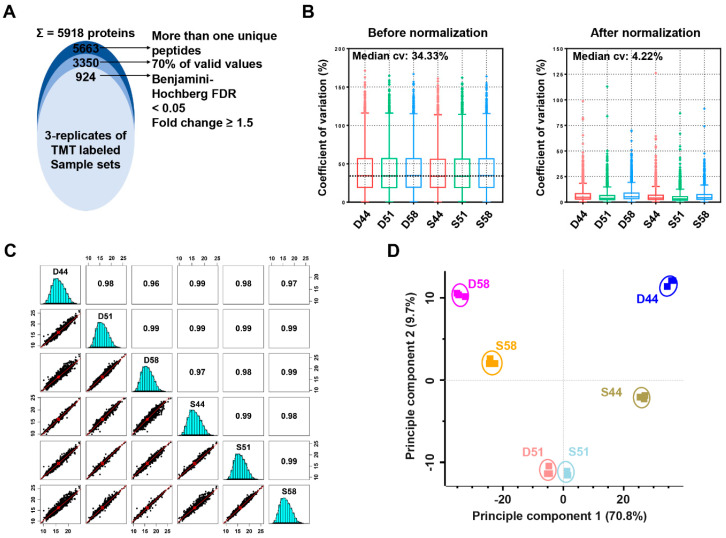
Proteomic analysis of filling stage seed samples by TMT based quantitative approach. (**A**) Venn diagram showing the distribution of total identified and significantly modulated proteins followed by a narrow-down approach among samples. (**B**) The coefficient of variation (CV) values showing the improvement of the quantitative reproducibility of all proteins normalized by the internal reference scaling (IRS) method using boxplot. The median CV values of all the samples decreased from 34.3% to 4.3% following the IRS normalization approach. (**C**) Multi-scatter plot showing the reproducibility across three replicates of each sample reveals with Pearson correlation value. (**D**) Principle component analysis showing clear separation of 924 significantly modulated proteins.

**Figure 6 cells-09-01517-f006:**
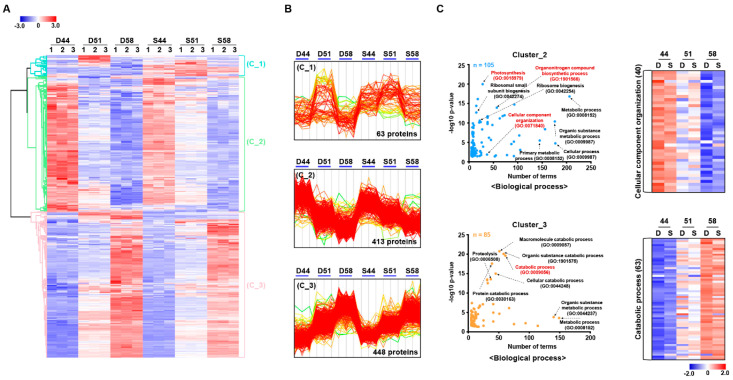
Hierarchical clustering analysis (HCA) clustering and gene ontology (GO) enrichment analysis of the differentially modulated proteins of seed filling stage samples. Heatmap (**A**) and abundance profile (**B**) showing the distribution of 924 significantly modulated proteins were divided into three major clusters based on their abundance among six samples. (**C**) The biological process of GO analysis shown distinct terms in each cluster mainly involved in the cellular component organization in cluster_2 and catabolic process in cluter_3.

**Figure 7 cells-09-01517-f007:**
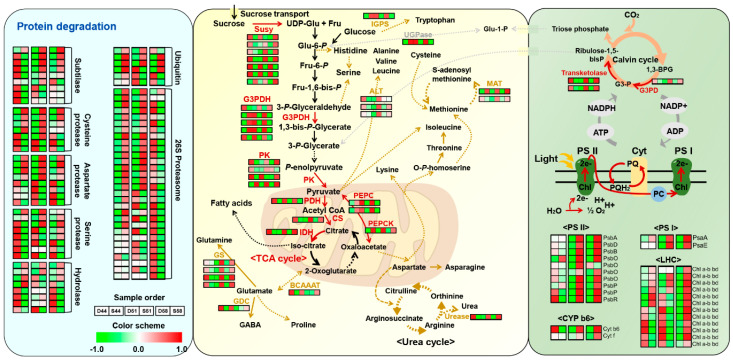
Schematic mapping of identified proteins to visualize the overall representative changes when comparison carried out Daewon versus Saedanbaek varieties during seed filling stages. The abundance differences of each protein are depicted by red and green color scheme. Abbreviations used: SuSy (sucrose synthase); GAPDH (glyceraldehyde 3-phosphate dehydrogenase); PK (pyruvate kinase); PDH (pyruvate dehydrogenase); CS (citrate synthase); IDH (isocitrate dehydrogenase); PEPC (phosphoenolpyruvate carboxylase); PEPCK (phosphoenolpyruvate carboxylkinase); IGPS (indole-3-glycerol phosphate synthase); UGPase (UTP-glucose pyrophosphorylase); ALT (alanine transaminase); MAT (methionine adenosyltransferase); BCAAT (branched-chain amino acid aminotransferase).

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
