# Peer review of "In-Depth Investigation of Low-Abundance Proteins in Matured and Filling Stages Seeds of Glycine max Employing a Combination of Protamine Sulfate Precipitation and TMT-Based Quantitative Proteomic Analysis"

_cells, 2020, doi:10.3390/cells9061517_

Round 1

Reviewer 1 Report

The work reports the high resolution analysis of the proteome of soybean seeds  using pre-fractionation and TMT-based quantitative proteomics. The approach led to identification of a large amount of proteins in the filling stages seeds of two soybean cultivars. Functional annotation of the differential proteins revealed differences in  major metabolic processes between the two cultivars. The results bring new insights into proteome changes during filling stages in soybean seeds, and provide a methodology for a dissection of seed proteome including low-abundance proteins that are generally missed with other methods.

The work is well performed with state-of-the-art techniques and is well written except for few mistyping errors and periods that are sometimes too long and not easy to read.

Minor corrections:

Line 53: These HAPs depletion methods seem to be effective and broad applied for the depletion of HAPs from various other plants including broad bean, pea, wild soybean, and peanut which contain their own specific HAPs.

Change to: These HAPs depletion methods seem to be effective and are broadly applied for the depletion of HAPs from different plants including broad bean, pea, wild soybean and peanut, which contain their own specific HAPs.

Line 69: Thus, the tandem mass tags (TMTs) were presented as an alternative approach which have the same features with other isotope labeling techniques where all the mass tags have identical nominal mass, chemical structure, and allowed up to 16 samples per MS run.

Change to: Thus, the tandem mass tags (TMTs) were presented as an alternative approach having the same features of other isotope labeling techniques, with mass tags of identical nominal mass and chemical structure, and that allowed up to 16 samples per MS run.

Line 136 and 138: 15,922 g 

Change to: 15,922 x g

Line 215-220: the period is too long, in the sentence "using calculating by mean from sum value of each  column and second internal reference scaling (IRS) method with trimmed mean of M values (TMM)  normalization was applied among the technical replicates.." punctuation is missing.

Line 228-332: the period is too long and twisted, split in shorter sentences.

Author Response

* Reviewer #1: The work reports the high resolution analysis of the proteome of soybean seeds using pre-fractionation and TMT-based quantitative proteomics. The approach led to identification of a large amount of proteins in the filling stages seeds of two soybean cultivars. Functional annotation of the differential proteins revealed differences in major metabolic processes between the two cultivars. The results bring new insights into proteome changes during filling stages in soybean seeds, and provide a methodology for a dissection of seed proteome including low-abundance proteins that are generally missed with other methods.

The work is well performed with state-of-the-art techniques and is well written except for few mistyping errors and periods that are sometimes too long and not easy to read.

- Reply: Thank you for your positive and constructive comments on our manuscript.

-Minor corrections:

  1. Line 53: These HAPs depletion methods seem to be effective and broad applied for the depletion of HAPs from various other plants including broad bean, pea, wild soybean, and peanut which contain their own specific HAPs.

Change to: These HAPs depletion methods seem to be effective and are broadly applied for the depletion of HAPs from different plants including broad bean, pea, wild soybean and peanut, which contain their own specific HAPs.

- Reply: Thank you. We have modified the sentence as per the suggestion (Lines 53-55).

  1. Line 69: Thus, the tandem mass tags (TMTs) were presented as an alternative approach which have the same features with other isotope labeling techniques where all the mass tags have identical nominal mass, chemical structure, and allowed up to 16 samples per MS run.

Change to: Thus, the tandem mass tags (TMTs) were presented as an alternative approach having the same features of other isotope labeling techniques, with mass tags of identical nominal mass and chemical structure, and that allowed up to 16 samples per MS run.

- Reply: We have modified the sentence as per the suggestion. Thank you (Lines 69-72).

  1. Line 136 and 138: 15,922 g

Change to: 15,922 x g

- Reply: Checked and corrected as per the suggestion.

  1. Line 215-220: the period is too long, in the sentence "using calculating by mean from sum value of each column and second internal reference scaling (IRS) method with trimmed mean of M values (TMM) normalization was applied among the technical replicates.." punctuation is missing.

- Reply: Thank you for your suggestion. We have modified the sentence as per the reviewers’ suggestion (Lines 214-219).

  1. Line 228-332: the period is too long and twisted, split in shorter sentences.

- Reply: We have reframed the sentences following reviewer’s comments to make those clearer to the readers (Lines 228-363).

Reviewer 2 Report

The manuscript entitled “In-depth investigation of low-abundance proteins in matured and filling stages seeds of Glycine max employing a combination of protamine sulfate precipitation and TMT-based quantitative proteomic analysis” provides novel insight about soybean content of proteins by a combination of methods. Indeed, results showed the efficacy of the methods and the manuscript is well written and presented.

I have only two suggestions which are:

  1. In the last paragraph of introduction, it would be better if you would state in a more appropriate way the current scopes of this manuscript, so please rewrite from line 78 – 84. and
  2. Although discussion is really well structured, I would suggest to provide a conclusion paragraph in the end or concluding remarks.

Author Response

* Reviewer #2: The manuscript entitled “In-depth investigation of low-abundance proteins in matured and filling stages seeds of Glycine max employing a combination of protamine sulfate precipitation and TMT-based quantitative proteomic analysis” provides novel insight about soybean content of proteins by a combination of methods. Indeed, results showed the efficacy of the methods and the manuscript is well written and presented.

- Reply: Thank you for your positive and constructive comments on our manuscript.

I have only two suggestions which are:

  1. In the last paragraph of introduction, it would be better if you would state in a more appropriate way the current scopes of this manuscript, so please rewrite from line 78 – 84.

- Reply: Thank you for your suggestion. We have modified the paragraph as per the suggestion (Lines 79-90).

  1. Although discussion is really well structured, I would suggest to provide a conclusion paragraph in the end or concluding remarks.

- Reply: We have added the concluding remark as per the suggestion (Lines 600-614).